# Neck Muscle Vibration Alters Cerebellar Processing Associated with Motor Skill Acquisition of a Proprioceptive-Based Task

**DOI:** 10.3390/brainsci13101412

**Published:** 2023-10-04

**Authors:** Hailey Tabbert, Ushani Ambalavanar, Bernadette Murphy

**Affiliations:** Faculty of Health Sciences, Ontario Tech University, Oshawa, ON L1G 0C5, Canada; hailey.tabbert@ontariotechu.net (H.T.); ushani.ambalavanar@ontariotechu.net (U.A.)

**Keywords:** neck muscle vibration, cerebellar processing, motor learning, body schema, somatosensory evoked potentials

## Abstract

Experimentally induced neck fatigue and neck pain have been shown to impact cortico-cerebellar processing and sensorimotor integration, assessed using a motor learning paradigm. Vibration specifically impacts muscle spindle feedback, yet it is unknown whether transient alterations in neck sensory input from vibration impact these neural processing changes following the acquisition of a proprioceptive-based task. Twenty-five right-handed participants had electrical stimulation over the right median nerve to elicit short- and middle-latency somatosensory evoked potentials (SEPs) pre- and post-acquisition of a force matching tracking task. Following the pre-acquisition phase, controls (CONT, *n* = 13, 6 F) received 10 min of rest and the vibration group (VIB, *n* = 12, 6 F) received 10 min of 60 Hz vibration on the right sternocleidomastoid and left cervical extensors. Task performance was measured 24 h later to assess retention. Significant time by group interactions occurred for the N18 SEP peak, 21.77% decrease in VIB compared to 58.74% increase in CONT (F_(1,23)_ = 6.475, *p* = 0.018, n_p_^2^ = 0.220), and the N24 SEP peak, 16.31% increase in VIB compared to 14.05% decrease in CONT (F_(1,23)_ = 5.787, *p* = 0.025, n_p_^2^ = 0.201). Both groups demonstrated improvements in motor performance post-acquisition (F_(1,23)_ = 52.812, *p* < 0.001, n_p_^2^ = 0.697) and at retention (F_(1,23)_ = 35.546, *p* < 0.001, n_p_^2^ = 0.607). Group-dependent changes in the SEP peaks associated with cerebellar input (N18) and cerebellar processing (N24) suggests that an altered proprioceptive input from neck vibration impacts cerebellar pathways.

## 1. Introduction

Workers in several occupations encounter vibration on a daily basis presented by handheld tools (hand-transmitted vibration (HTV)) or from operating large equipment (whole body vibration (WBV)) [1]. Many occupational tasks require precise upper limb control and often call upon workers to master novel motor skills while being exposed to vibration. This type of occupational vibration exposure has been associated with increased susceptibility to musculoskeletal pain in the back, neck, hips, and upper limbs [1]. 

In order to constantly guide and monitor movements of the body, the brain constructs an internal map of the body’s position in reference to other body parts and the objects surrounding it, known as body schema [2]. The central nervous system (CNS) monitors and modifies the body schema based on proprioceptive feedback and previous body movements, relying on both feedforward and feedback models [3,4,5,6]. Sensorimotor integration (SMI) relies on feedforward and feedback networks to integrate somatosensory information to generate an appropriate motor response, which is vital to motor control and the acquisition of motor-based tasks [7]. Changes in SMI are driven by cortico-cerebellar networks due to their involvement during the early stages of motor skill acquisition through synaptic modification of the strength of parallel fiber inputs projecting to the cerebellum via climbing fibers [8,9]. This directly influences the activity in neural pathways between the cerebellum and sensorimotor cortex [10,11,12].

Chronic or transient changes in neck sensory input from pain, postural stress, or fatigue has been suggested to influence processes pertaining to SMI [13,14], likely due to the processing of altered afferent input. This has been seen as impaired upper limb performance accuracy [15,16], proprioception [17,18,19,20,21,22], and sensorimotor adaptation with respect to motor acquisition both neuronally and behaviorally [13,23,24]. Neuronal changes in response to motor skill acquisition have been quantified using somatosensory evoked potentials (SEPs), demonstrating changes in early (short-latency) SEP peaks related to SMI [14,25,26,27,28]. Somatosensory evoked potentials (SEPs) are complex waveforms that are generated in response to the external somatosensory stimulation of the peripheral nerve of interest [10] as they pass along neural generators via the dorsal column medial lemniscus pathway. Healthy participants who underwent cervical extensor muscle (CEM) fatigue prior to the acquisition of a motor tracking task had significant increases in the N24 (cerebellar–somatosensory processing) and N30 SEP peak (SMI) amplitudes when compared to the controls [27]. This suggests that experimentally induced alterations in neck sensory input impacts cerebellar processes as well as neural correlates (motor circuit of the cortico-basal ganglia–thalamo-cortical loop) pertaining to SMI, whereas the individuals with recurrent neck pain who acquired a pursuit motor task had significant increases in the N18 (inhibitory activity along the olivary–cerebellar pathway) and the N24 compared to healthy controls, with no differential changes in the N30 [13]. This was coupled with reduced performance accuracy following the acquisition and retention of a motor tracing task [13]. The findings of that study suggest that chronic alterations in neck sensory input give rise to maladaptive changes with respect to cortico-cerebellar networks, suggesting that sensory feedback is essential for body schema, and, in turn, sensorimotor coordination. The observed changes in both studies could be the result of altered sensory feedback from muscle spindles or altered muscle spindle firing since the upper neck muscles are densely populated with muscle spindles. Given that muscle spindles are a major source of proprioceptive feedback, altered spindle feedback may lead to altered cortico-cerebellar processing [29,30,31,32], while altered spindle firing could possibly impact neck muscle tone [33] and biomechanical connections between the neck and upper limb [27,34].

The type of muscle spindle feedback and afferent feedback that occurs in response to experimentally induced neck muscle fatigue or neck pain differs. Altered muscle spindle firing may occur alongside changes in muscle properties in response to pain [35,36] and may persist whether there is pain on the day of testing or not. The effects of experimentally induced neck fatigue can be short-lasting, with recovery occurring approximately 5 min following fatigue protocols [18,37], making it difficult to infer the mechanism by which altered neck inputs affect upper limb motor learning. It is therefore of interest to examine the effects of altered neck muscle spindle input on upper limb SMI without the confounding effects of pain. As such, muscle vibration protocols have been utilized to examine changes in sensorimotor function [22,38,39,40]. The short-term, experimentally induced alteration of input from the muscle spindles can be elicited using high-frequency, low-amplitude vibration over the muscle belly of interest, enabling excitation of the primary (Ia) afferents without causing pain or discomfort within the musculature [41,42,43]. A vibration frequency between 30 and 100 Hz is sufficient to induce long-lasting physiological effects [42,43]. A vibration frequency that exceeds 30 Hz is perceived by the CNS as joint rotation and movement, thereby inducing illusions of movement [40,42]. Neck vibration, at frequencies between 60 Hz and 100 Hz, has been shown to impair upper limb proprioception of the elbow and forearm [22,39,43], suggesting that neck muscle vibration may impact SMI, since proprioceptive feedback is needed for the underlying processes of SMI. 

Recent work found that 10 min of 60 Hz vibration over the right sternocleidomastoid and left CEM muscles was sufficient to induce impairments in upper limb proprioception [39]. This suggests that proprioceptive-based motor task performance may also be impacted by neck vibration. As past work has demonstrated that either sensorimotor training [44] or the acquisition of a force-matching task induces functional changes in cortico-cerebellar networks [44,45] or increases the excitability of descending motor pathways [46] or hemispheric and lateralized specialization in areas of the brain involved in proprioceptive processing and movement planning, alongside improvements in motor performance following the acquisition and retention of the task [45,47]. The one study that investigated differential changes in short-latency SEP peaks following the acquisition of a proprioceptive-based tracking task compared to the acquisition of a motor tracing task in healthy controls found neurophysiological differences in the way the CNS responded to visuomotor information versus a task heavily reliant on proprioceptive input [25]. This study found a decrease in SEP peaks associated with somatosensory processing and an increase in N18 [25], suggesting that sensory feedback gating alongside greater inhibitory activity in cortico-cerebellar networks was required to accurately produce and modulate forces. This in correspondence with the known effects of neck muscle vibration on afferent feedback mean that it is possible that neck muscle vibration might impact an individual’s ability to acquire a proprioceptive-based motor task and perform it accurately, either post-acquisition or following memory consolidation (at retention). To our knowledge, there are no studies that have examined the impact of altered neck sensory input via vibration on the acquisition of a force-matching tracking task. 

The purpose of this study was to determine the effects of neck muscle vibration on the neurophysiological response to the acquisition of a novel force matching tracking task (FMTT) by examining early- and middle-latency SEP peaks and motor performance. It was hypothesized that neck muscle vibration would lead to differential changes in SEP peaks associated with somatosensory processing, cerebellar pathways, and sensorimotor integration. A secondary hypothesis was that the vibration group would demonstrate poorer force tracking accuracy and altered motor learning patterns compared to the control group. 

## 2. Materials and Methods

### 2.1. Participants

In total, 25 healthy, right-handed participants, 13 males and 12 females (average age: 22.03 ± 2.62 years), were recruited and randomly allocated to the vibration (*n* = 12, 6 females, aged 21.32 ± 1.97) or control (*n* = 13, 6 females, aged 22.75 ± 3.17) groups. Inclusion criteria for the study required all participants to be between the ages of 18 and 35 years old and be right hand dominant, determined by a score of above 40 on the Edinburgh handedness inventory. The participants could not have recurrent or chronic neck pain, indicated by a score of less than 5 on the Neck Disability Index [48]. Participants who had any of the following conditions were excluded as they may alter electroencephalography (EEG) suitability and/or impact central processing, such as: multiple sclerosis, epilepsy, seizure disorders, recurrent neck pain, autism spectrum disorder (ASD), and attention deficit hyperactivity disorder (ADHD) [13,23,28,35,49,50,51]. This research was reviewed by the University of Ontario Institute of Technology (Ontario Tech University) Research Ethics Board and received ethical approval [REB #16520]. Written informed consent was received from every participant.

### 2.2. SEP Stimulation Protocol and Recording Parameters

The right median nerve was stimulated at the wrist using conductive adhesive hydrogel electromyography (EMG) electrodes (MEDITRACE^TM^ 130, Kendall, Mansfield, MA, USA). The surface EMG electrodes were placed 2–3 cm proximal to the distal crease of the wrist, with the anode proximal to the wrist [52]. Constant wave pulses of 0.2 ms were administered. A stimulation intensity that exceeded the motor threshold of the abductor pollicis brevis (APB) muscle was used, i.e., visible twitch in the APB muscle. SEP stimulation was conducted at 2 frequencies to allow for the clear identification of SEP peaks. The median nerve was stimulated at a frequency of 2.47 Hz to enable optimal recordings of the N30 peak, followed by a stimulation frequency of 4.98 Hz to produce a clear N24 peak through attenuation of the N30 peak [53]. 

Peripheral SEP peaks were recording using EMG electrodes on Erb’s point over the brachial plexus referenced to an ipsilateral ear clip, and over the spinous process of the 5th cervical vertebrae with reference electrodes placed on the anterior tracheal cartilage and contralateral clavicle [52]. Cortical SEP peaks were recording using a 64 lead ANT Neuro Waveguard EEG cap (ANT Neuro, the Netherlands, Manufactured by Eemagine, Berlin, Germany). The EEG cap was fitted for each participant using the internationally standardized 10–20 system in accordance with the IFCN guidelines [52]. Each cortical electrode was filled with conductive gel to reduce electrical impedance below 10 Ω and improve signal acquisition. SEP peaks were recorded at a sampling frequency of 2048 Hz. In total, 1000 sweeps were recorded and averaged at each stimulation frequency.

### 2.3. Neck Muscle Vibration

Neck muscle vibration was applied using two DC-motor vibrators measuring 4 cm in diameter. The vibrator heads were placed antero-laterally, 2 cm from the midline and 5 cm below the mastoid for the sternocleidomastoid (SCM) and 2–3 cm lateral to the C5 spinous process for the CEM [43]. The vibrators were firmly affixed to the neck using hypafix tape to ensure sufficient contact was maintained. Participants in both vibration groups had high-frequency, low-amplitude vibration applied to the right SCM and left CEM at a frequency of 60 Hz for 10 min [39]. Those in the control group were fitted with the vibrators which were not turned on for the same duration. Participants were fitted with blackout goggles for the duration of the vibration protocol to eliminate visual feedback. 

### 2.4. Force Matching Tracking Task 

Participants were seated in a chair with their right arm fixed to a height adjustable table housing a small 50 kg force transducer that fit comfortably against their right thumb. A computer monitor was placed 70 cm in front of them displaying visual information regarding the task. Prior to beginning the task, signal noise was eliminated from the force output of the transducer. An average of 3–5 maximal voluntary contractions (MVCs) of the APB muscle were taken to calibrate the target line to the strength of each individual participant. The FMTT required participants to push against the force transducer using their right thumb to match a series of white-dotted square sinusoidal waves calibrated to their thumb strength, 2 to 12% of their average MVC [25]. This range was chosen to ensure that thumb muscle fatigue did not occur during motor acquisition. A set of red error bars were situated 5% above and below the white-dotted target line as a guide for the participants. Pressing harder would direct the trace line upwards, while pressing lightly would direct the trace line downwards. Targets were presented on the monitor as one trial with two 10 s long force traces. Target lines and tracking performance were displayed in real-time using a custom made LABVIEW software program (National Instruments, Austin, TX, USA). Prior to beginning the task, the participants took part in one familiarization trial. The FMTT consisted of 5 phases, each containing 4 blocks with 12 trials. The phases included one pre-acquisition phase, 3 acquisition phases, one post-acquisition phase, and one retention phase. Phases of the task were delivered in sequence, while the blocks within each phase were randomized for each participant. Each trial lasted 22 s in duration with a total of 72 trials across all phases.

### 2.5. Experimental Procedure

The participants completed baseline dual SEP recordings. The participants then completed the pre-acquisition phase of the FMTT. The participants in the vibration group received 10 min of neck muscle vibration while the controls received 10 min of rest. Following vibration or rest, the participants completed the acquisition and post-acquisition phases of the FMTT followed by the post-intervention dual SEP recordings. Then, 24 h later, the participants returned to complete the retention phase of the FMTT. The flow of the experiment is provided in Figure 1.

### 2.6. Data Processing

EEG signals were amplified using a TMSi REFA-8 amplifier (TMSi, Oldenzaal, the Netherlands) with a gain of 10,000 during post-processing in Advanced Source Analysis (ASA™, ANT Neuro, Hengelo, the Netherlands). Cortical SEPs were extracted following the removal of eye blinks and artifacts (e.g., ocular movement and muscle activity) with a detection parameter set at a minimum of −100 μv and a maximum of +100 μv. Following this step, a band-pass filter with a low cut-off of 0.2 Hz and a high cut-off of 1000 Hz was applied to the EEG signal [13]. The peripheral signals recorded using Signal^®^ software (Cambridge Electronic Design, Cambridge, UK) were amplified by a gain of 10,000 during pre-processing. In total, 1000 sweeps were averaged to enable the identification of SEP peaks [52]. The peripheral N9, N11, and N13 SEP peaks were analyzed using Signal^®^ Software (Cambridge Electronic Design Limited, Cambridge, UK). Cortical SEP Peaks (N18, N20, N24, P25, N30, and N60 SEP) were analyzed using ASA^TM^ Software (version 4.10.1). Amplitudes were measured at the peak of interest in accordance with the IFCN guidelines [52]. Changes in SEP amplitudes from pre-intervention to post-intervention were reported as a percentage increase or decrease from the baseline for each group.

FMTT data were recorded and analyzed using a customized LabVIEW data analysis program. A 0.5 s moving average was conducted to smooth the force trace data prior to analysis. Tracking accuracy was measured as the absolute percent error calculated as the difference between the participant’s trace line and the presented target line [25]. Absolute percent error and standard deviation of the error were calculated for each block of the baseline, post-acquisition, and retention phases separately. Data were normalized to the baseline by dividing the phase average by the baseline average for each phase and then averaged for each group. Single subject data sets for all SEP and motor performance data are attached as Appendix A.

### 2.7. Statistical Analysis

The Shapiro–Wilk test was used to test for a normal distribution for all datasets. If violated, log transformations were applied to ensure that data were normally distributed. Mauchly’s test of sphericity was used to test sphericity for the performance accuracy data. Greenhouse–Geisser corrections were reported for performance accuracy data that violated Mauchly’s test of sphericity. Statistical significance was set as *p* ≤ 0.05 for all statistical tests. All statistical analyses were performed using SPSS version 26 (Armonk, New York, NY, USA).

Normalized SEP peak data were analyzed using a two-way repeated measures analysis of variance (ANOVA) with group (control vs. vibration) as a factor and time (pre/post FMTT) as the repeated measure. The Benjamini–Hochberg test was used to correct for multiple comparisons that are independent of each other as each SEP peak has its own set of neural generators. This correction controls for the likelihood of type I error or false discovery rate by ranking the individual *p*-value from smallest to largest and is then compared to the Benjamini–Hochberg critical value [54]. With this correction, datasets are considered statistically significant if the adjusted *p*-value is smaller than the chosen family-wise error rate. The false discovery rate was set at 0.2 in an excel spreadsheet created to test for Benjamini–Hochberg [54]. The *p*-values provided in the Section 3 are the unadjusted *p*-values, as recommended [54]; however, statistical significance for SEP peak data was determined by the Benjamini–Hochberg test.

A 2 × 3 mixed methods repeated measures ANOVA was used to compare the mean difference in the performance accuracy with between-subject and within-subject factors, with group (control vs. vibration) and time (pre-acquisition, post-acquisition, and retention of FMTT) as the repeated measure. Pre-planned repeated contrasts were included in the FMTT repeated measures ANOVA as this permits for the pairwise comparison of pre-acquisition and post-acquisition, as well as post-acquisition and retention [55]. 

Then, 95% confidence intervals were calculated and reported for all measures. Partial eta squared values are reported with 0.01 equal to small, 0.06 equal to medium, and 0.14 equal to large effect sizes for the ANOVAs [56]. 

## 3. Results

### 3.1. SEP Peak Amplitudes

All participants met the N9 criteria of a change within ± 20%; therefore, all SEP data were included in the analysis [52]. There was no effect of time (F_(1,23)_ = 0.012, *p* = 0.915, n_p_^2^ = 0.001, observed power _(1−β)_ = 0.051) or time by group interaction (F_(1,23)_ = 2.585, *p* = 0.121, n_p_^2^ = 0.101, observed power _(1−β)_ = 0.338) for the N9. Consistency of the N9 SEP peak pre or post is critical to ensure that subsequent changes in spinal and cortical SEP peaks are the result of changes in neural activity following motor learning and experimental manipulation. Table 1 shows the results of each SEP peak that was analyzed.

N18 SEP Peak: There was a significant time by group interaction (F_(1,23)_ = 6.475, *p* = 0.018, n_p_^2^ = 0.220, observed power _(1−β)_ = 0.642) where the amplitude decreased by 21.77% in vibration and increased by 58.74% in the controls (Figure 2a). 

N24 SEP Peak: There was a significant time by group interaction (F_(1,23)_ = 5.787, *p* = 0.025, n_p_^2^ = 0.201, observed power _(1−β)_ = 0.575) where the SEP peak amplitude increased by 16.31% in vibration and decreased by 14.05% in the controls (Figure 2b). Figure 3 shows a representative dataset of the average N18 and N24 waveform, pre- and post-acquisition in both groups.

N11 SEP Peak: There was a significant effect of time (F_(1,23)_ = 446.748, *p* < 0.001, n_p_^2^ = 0.953, observed power _(1−β)_ = 1.00), with the SEP peak amplitude increasing by 18% in the controls and decreasing by 4% in vibration from pre- to post-acquisition. The time by group interaction was not significant (F_(1,23)_ = 0.166, *p* = 0.688, n_p_^2^ = 0.007, observed power _(1−β)_ = 0.068).

N13 SEP Peak: There was no effect of time (F_(1,23)_ = 0.020, *p* = 0.920, n_p_^2^ = 0.00, observed power _(1−β)_ = 0.051) or time by group interactions (F_(1,23)_ = 0.00, *p* = 0.984, n_p_^2^ = 0.00, observed power _(1−β)_ = 0.050) for the changes observed in the N13 SEP peak.

N20 SEP Peak: There were no significant effects of time (F_(1,23)_ = 2.969, *p* = 0.099, n_p_^2^ = 0.119, observed power _(1−β)_ = 0.378) or time by group interactions (F_(1,23)_ = 2.299, *p* = 0.144, n_p_^2^ = 0.095, observed power _(1−β)_ = 0.306) for the trends observed in the N20 peak amplitude.

P25 SEP Peak: There was a significant effect of time (F_(1,23)_ = 4.556, *p* = 0.044, n_p_^2^ = 0.165, observed power _(1−β)_ = 0.556) as the SEP peak amplitude increased for both groups, by 18.55% in the vibration and 9% in the control groups (Figure 4a). The time by group interaction for this peak was not significant (F_(1,23)_ = 0.727, *p* = 0.403, n_p_^2^ = 0.032, observed power _(1−β)_ = 0.129).

N30 SEP Peak: There was a significant effect of time (F_(1,23)_ = 4.403, *p* = 0.047, n_p_^2^ = 0.161, observed power _(1−β)_ = 0.513) which increased for both groups, by 10.96% in the vibration and 7.38% in the control groups (Figure 4b). There was no significance for time by group interaction (F_(1,23)_ = 0.216, *p* = 0.647, n_p_^2^ = 0.010, observed power _(1−β)_ = 0.073).

N60 SEP Peak: There were no effects of time (F_(1,23)_ = 0.740, *p* = 0.399, n_p_^2^ = 0.033, observed power _(1−β)_ = 0.130) or group (F_(1,23)_ = 0.550, *p* = 0.466, n_p_^2^ = 0.024, observed power _(1−β)_ = 0.109) on the N60 SEP peak.

### 3.2. Motor Performance Accuracy

Absolute motor performance error had a main effect of time (F_(1,23)_ = 32.326, *p* < 0.001, n_p_^2^ = 0.584, observed power _(1−β)_ = 1.00). The pre-acquisition to post-acquisition contrast had a significant effect of time (F_(1,23)_ = 40.092, *p* < 0.001, n_p_^2^ = 0.635, observed power _(1−β)_ = 1.00), where vibration improved by 14.22% and the controls improved by 12.3%. The post-acquisition to retention contrast was not a significant for time (F_(1,23)_ = 0.002, *p* = 0.964, n_p_^2^ = 0.000, observed power _(1−β)_ = 0.050) or time by group (F_(1,23)_ = 0.303, *p* = 0.587, n_p_^2^ = 0.013, observed power _(1−β)_ = 0.082), where vibration deteriorated by 0.59% and the controls improved by 0.48%. The absolute Sd of error also had a main effect in terms of time (F_(1,23)_ = 24.033, *p* < 0.001, n_p_^2^ = 0.511, observed power _(1−β)_ = 1.00). The pre-acquisition to post-acquisition contrast had a significant effect in terms of time (F_(1,23)_ = 40.953, *p* < 0.001, n_p_^2^ = 0.640, observed power _(1−β)_ = 1.00), where variance decreased by 11.45% in the controls and by 16.46% in vibration (see Table 2).

Normalized motor performance error had a main effect in terms of time (F_(1,23)_ = 38.240, *p* < 0.001, n_p_^2^ = 0.624, observed power _(1−β)_ = 1.00). The pre-acquisition to post-acquisition contrast had a significant effect in terms of time (F_(1,23)_ = 52.277, *p* < 0.001, n_p_^2^ = 0.694, observed power _(1−β)_ = 1.00), where vibration improved by 12.71% and the controls improved by 11.87%. The post-acquisition to retention contrast was not a significant for time (F_(1,23)_ = 0.208, *p* = 0.652, n_p_^2^ = 0.009, observed power _(1−β)_ = 0.072) or time by group (F_(1, 23)_ = 0.411, *p* = 0.528, n_p_^2^ = 0.018, observed power _(1−β)_ = 0.094), where vibration deteriorated by 0.88% and the controls improved by 0.17%. (see Figure 5). Normalized SD of error also had a main effect in terms of time (F_(1,23)_ = 17.014, *p* < 0.001, n_p_^2^ = 0.425, observed power _(1−β)_ = 1.00). The pre-acquisition to post-acquisition contrast had a significant effect in terms of time (F_(1,23)_ = 26.036, *p* < 0.001, n_p_^2^ = 0.531, observed power _(1−β)_ = 0.998), where variance decreased by 9% in the controls and 12% in vibration (see Figure 6).

## 4. Discussion

This is the first study to assess the impact of high-frequency, low-amplitude vibrations on the neck musculature using a motor learning paradigm with a proprioceptive-based motor task, FMTT, via neurophysiological changes, i.e., SEP peaks and motor performance accuracy. This study revealed that 60Hz vibration over the right SCM and left CEM for a duration of 10 minutes leads to differential changes in SEP peaks associated with cerebellar input (N18) and cerebellar–somatosensory processing (N24). Both groups showed similar increases in SEP peaks associated with somatosensory processing (P25) and sensorimotor integration (N30), seen as an effect of time, whereas the SEP associated with ascending volley arriving at the spinal cord (N11) increased in the control group with a small decrease in the vibration group. The vibration and control group demonstrated significant improvements in motor performance following the acquisition of the FMTT, reflected by an effect of time in error and variability in error. The differential changes in cerebellar pathways indicates that neck muscle vibration at 60Hz for 10 minutes was sufficient to impair proprioceptive input via the primary afferent muscle spindles, which then impacted body schema, as indicated by the differential changes in specific SEP peaks.

### 4.1. SEPs

Previous work found that when SEPs were measures in a control experiment that did not include motor learning (use of a mental recitation task instead), there were no changes in SEP peaks [57]. This suggests that vibration alone was not sufficient to alter early SEP peaks, providing strong evidence that the changes observed in experiment one are due to the effects of vibration on motor learning and not the effects of vibration on SEPs.

The N18 SEP peak originates in the inferior olive, dorsal column medial lemniscus (DCML) nuclei of the lower medulla, and the midbrain–pontine region [58]. This peak reflects inhibitory activity at the level of the dorsal column nuclei, possibly due to the collaterals from the cuneate nucleus [58,59]. Changes in inferior olive activity are associated with the performance of well-learned movements, and inferior olive activity is known to increase during motor acts, contributing to online motor control and motor learning [9]. This is likely due to increased climbing fiber input from the inferior olive to the cerebellum to minimize errors [9,60]. A novel finding from this study indicated that those in the vibration group experienced a reduction in N18 amplitude following motor acquisition, which may reflect the reduced inhibition of olivary–cerebellar inputs or the selective filtering of cerebellar inputs at the level of the inferior olive. This was likely necessary to continuously refine motor output to the right thumb, potentially due to alterations in proprioceptive inputs from the neck due to vibration. This suggests that those in the vibration group had a decreased reliance on proprioceptive input when learning the motor acquisition task compared to controls. This could be because the vibration-induced alterations in the body schema of the upper limb in relation to the neck made proprioceptive feedback less reliable. It is also possible that vibration may have impacted the flexibility of the CNS to respond through the freezing of various motor systems to achieve the motor skill, seen as decreased variability in the N18 SEP peak compared to the control group. Previous work utilizing a similar methodology saw an increased N18 amplitude in controls following the acquisition of a novel force matching task [25,28,61], supporting the findings from the current study which show that the controls experienced an increase in N18 amplitude post-acquisition. The transmission of sensory information from the neck, head, and upper limb is regulated by the cuneocerebellar tract, specifically the lateral cuneate nucleus located in the dorsolateral medulla at the level of the inferior olive [62]. The cuneate nuclei topographically relay precise proprioceptive information through complex feedback-regulated sensorimotor cerebellar connections to other areas of the cerebral cortex [58,63]. Therefore, changes in N18 amplitude could reflect alterations in cerebellar SMI and unconscious proprioception. 

The P25 peak reflects activity in Brodmann’s area one of the primary somatosensory cortex (S1) and is associated with the somatosensory processing of cutaneous inputs [64,65]. The increase in P25 amplitude seen in both groups suggests greater activity in S1 following the acquisition of the FMTT. This is in line with past work which illustrated an increased P25 following the acquisition of a motor pursuit task which was more heavily reliant on visuomotor integration compared to the current FMTT [13]. However, Ambalavanar et al. [25] reported a decreased P25 peak amplitude in healthy controls following the acquisition of a similar FMTT. Given this, it is likely that the 10 min rest period before motor acquisition in the controls played a role in somatosensory processing. The increase in the vibration group could be attributed to an increased reliance on sensory feedback to acquire the task. 

The N24 peak is generated near the wall of the central sulcus in the pathway linking the cerebellum and primary somatosensory cortex, reflective of cerebellar processing [10,11,52,66,67]. A novel finding in the current study was that the vibration group appeared to have altered cerebellar processing, as demonstrated by an increased N24 peak amplitude following motor learning. An increase in the N24 amplitude reflects increased cerebellar deep nuclei activity, as well as increased cerebellar to S1 processing and is indicative of a lack of disinhibition, or greater cerebellar inhibition in response to motor training. While this is the first study to report on quantifiable changes in cerebellar activity following vibration, others have utilized similar methodology to investigate N24 peak changes as a result of neck fatigue, pain, and joint dysfunction. Previous work has shown a similar increase in N24 amplitude in response to motor learning in groups that experience altered afferent input from the neck, including SCNP and fatigue [13,27]. The large decrease in the N24 peak amplitude seen in the controls is likely related to decreased cerebellar nuclei activity and decreased cerebellar processing subsequent to learning. During the early stages of motor acquisition, the cerebellar deep nuclei and cortico-cerebellar networks are highly active in order to contribute to motor adaptation and error correction as a skill is learned [68,69,70]. Although the deep nuclei contribute to the motor sequences used to execute a motor task, long-term representations of this sequence are stored elsewhere in the brain, resulting in an experience-dependent shift in cortical activation from the cerebellar cortex and deep nuclei when a skill is first established to cortico-striatal networks with extended practice [68,71]. A decreased N24 amplitude following motor learning is indicative of reaching the later stages of consolidated learning and forming a greater reliance on a well-formed internal schema [68]. The results from the current study suggest those in the vibration group were unable to reach the later stage of consolidated learning in a similar manner to the controls. The difference in N24 activity suggests alterations in motor learning at a cortical level even though this was not reflected in the motor performance data. This is likely due to the high sensitivity nature of SEPs, where the vibration was able to alter body schema sufficiently to induce neuroplasticity of the cerebellum but was not sufficient to impair motor performance.

The N30 peak is related to activity within complex connections linking the thalamus, basal ganglia, premotor, motor, and prefrontal cortices, which are all involved in SMI [10,65,72]. Both groups demonstrated similar increases in N30 peak amplitude post-acquisition, suggesting that increased activity in the neural correlates is involved with the SMI necessary for precise motor control and force modulation required during the FMTT. This aligns with the findings from previous work using similar methodology, which demonstrated a similar increase in N30 peak amplitudes in both experimental groups [25].

### 4.2. Motor Behavior

Motor learning can be assessed behaviorally by measuring improvements in motor performance and/or performance accuracy after the acquisition of a novel motor skill [73,74]. The consolidation of this skill is measured as a continued improvement in or maintenance of the skill 24 h following acquisition [70]. Memory consolidation occurs during sleep, which is why the retention of a motor skill is assessed after 24 h or so [45,75,76]. The increase in motor performance alongside improved accuracy from the baseline to post-acquisition in both groups is indicative of motor acquisition. Additionally, the maintenance of motor performance and level of accuracy from post-acquisition to retention suggests that both groups were able to retain the skill. These findings are supported by previous work using force-based motor learning paradigms which also reported improvements in motor performance following the acquisition and retention of the task [25,45,47]. Though significance was not shown when comparing motor performance error or variability between post-acquisition and retention for either group, trends at retention suggest that those in the vibration group may be regressing toward baseline performance, while the controls continued to maintain their post-acquisition accuracy. This might imply differences in the consolidation of motor skills following exposure to vibration and should be more closely investigated in future work. The lack of group differences in motor performance may be due to the challenge point framework, where the vibration group had to work harder to learn the task, overriding the impact of vibration-induced alterations in body schema [77]. Previous research has shown an inverted-U relationship between task difficulty and motor learning, where the second highest difficulty was most effective for skill acquisition while learning was delayed at the least and most difficult levels [78]. Another possible explanation could be the distinction between discrete and continuous motor tasks. The current task was discrete in nature, requiring distinct thumb movements that had a clear beginning and end which likely influenced the perceived task difficulty and attention demands. It is possible that a continuous version of the task may be more sensitive to group differences and may elicit greater learning effects following training. One final consideration is the effects of vibration on force production, which was required for the current motor learning task. While there is not any current work on neck vibration, previous work on WBV shows significant increases in peak force following short-term WBV [79], while others using concurrent WBV showed no effect on force production [80]. In contrast, focal muscle vibration programs lasting between 3 and 5 days lead to significant increases in peak torque, peak power, and endurance lasting up to 7 days after interventions for the muscle being vibrated [81,82]. While this is the first study to evaluate motor learning patterns following vibration, previous work has shown an increased tracking position error [22,43] as well as reduced precision and accuracy of the upper limb [39] following neck muscle vibration. Similar alterations in motor learning were observed in SCNP populations following the acquisition of a simple typing task [83] and the acquisition of a motor pursuit task [13].

### 4.3. Real-World Application

It is important to broaden our understanding of the effects of vibration on sensory processing and motor performance as employees in many occupational settings are exposed to high-frequency vibrations on a daily basis. Additionally, workers are often required to master novel motor skills and perform tasks requiring extreme precision while being exposed to vibration. This is especially common in dentistry where medical professionals use vibrating tools to perform procedures that require extensive precision and motor accuracy. Therefore, understanding the neurophysiological effects of vibration could provide valuable insights to workplace practices regarding work–rest ratios and limiting exposure. This work demonstrates that exposure to vibration generates long-lasting changes in cortical activity, specifically in the cerebellar processes involved in proprioception and motor learning. Though this work did not show differences in motor performance between groups, previous work has demonstrated marked changes in the accuracy and precision of the upper limb following vibration [39]. The changes in early SEP peaks are highly sensitive markers of altered neuroplasticity following vibration exposure, indicating that neck muscle vibration has effects on neuroplastic adaptations to subsequent motor learning. Given this, exposure to vibration has the potential to lead to errors which could impact the health and productivity of professionals, and in the case of medical occupations, it could also impact the well-being of patients. This work is important as it demonstrates that vibration induces acute alterations in spindle feedback, leading to changes in cortical processes associated with proprioception and motor acquisition that persist long after vibration exposure has ceased. Participants in the vibration group were able to learn the motor task because of the challenge point framework, but at what cost? It is likely that more attention was required to perform the task accurately, which would result in a lack of attention allocated to their work and other surroundings in a real-world setting. Future work should investigate the effects of vibration on more immersive and continuous motor learning paradigms as well as dual-task performance under multiple stimuli to further examine the relationship between vibration exposure and motor skill acquisition and how it translates to the workplace.

### 4.4. Limitations

The study sample consisted of university students ranging in age from 18 to 30; therefore, these results may not be generalizable to older adults or young children. It is possible that the participants were not blinded to the experimental condition as they were aware if their neck was being vibrated or not. However, this would be unlikely to have impacted the SEP peak changes as SEP peaks at 60 ms and below are “pre-cognitive”. Future work could consider applying the same vibration frequency for the same duration at another part of the body to control for blinding of the experimental condition. The motor learning paradigm employed in this work may not have been difficult enough to elicit differential behavioral changes. Future work should consider using a more complex task to allow for continuous learning. Subsequent studies could consider the assessment of movement (processes), as past work suggests that vibration-induced altered upper limb proprioception [39] would be likely to impact movement trajectories. 

## 5. Conclusions

Alterations in cerebellar processing were observed in the vibration group, even though the vibration group was still able to acquire and retain a novel force matching task. The SEP changes may reflect a greater reliance on proprioceptive feedback due to vibration-induced alterations in the proprioceptive inputs used to construct body schema. Future work should modify the force matching task to be more complex to allow for a continuous improvement to determine if the neurophysiological adaptations observed in the vibration group translate to more profound changes in motor performance and motor skill consolidation. In conclusion, this work demonstrates that neck muscle vibration induces alterations in cortical processing in the neural correlates associated with learning a proprioceptive-based motor paradigm.

## Figures and Tables

**Figure 1 brainsci-13-01412-f001:**
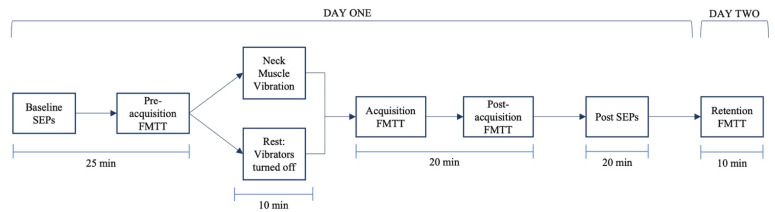
Flow of experimental procedure. Time durations for each procedure are labeled below each box.

**Figure 2 brainsci-13-01412-f002:**
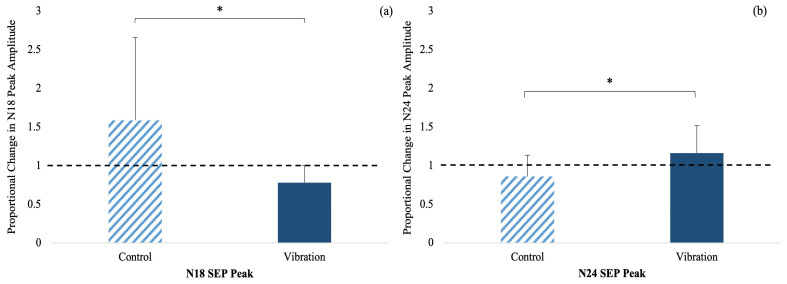
SEP peak amplitude changes with time by group interaction following motor acquisition of FMTT: (**a**) N18 peak amplitude and (**b**) N24 peak amplitude for all groups. Controls are dashed and vibration group are solid blue. Error bars represent SD. The black dotted line represents the normalized baseline amplitudes. (* *p* ≤ 0.05).

**Figure 3 brainsci-13-01412-f003:**
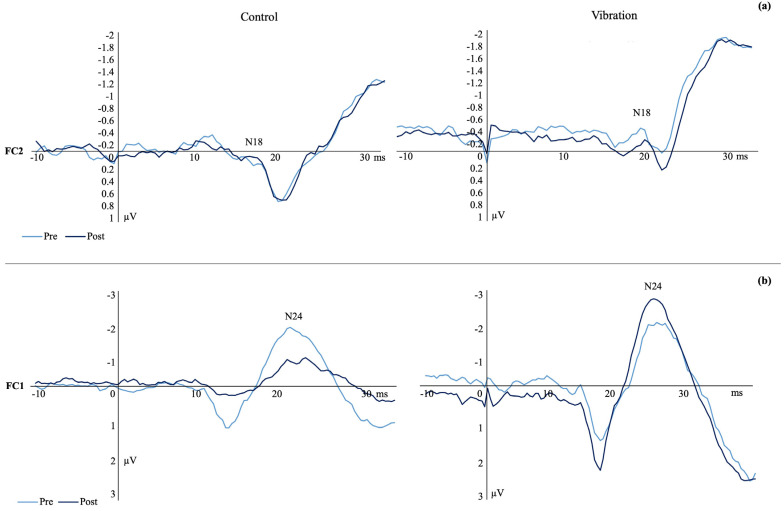
Representative datasets showing raw SEP peaks at baseline and post-acquisition for (**a**) N18 SEP peak amplitudes recorded from the FC2 electrode at a stimulation frequency of 2.4 Hz and (**b**) N24 SEP peak amplitude recorded from the FC1 electrode at a frequency of 4.98 Hz. Control (**left**) and vibration (**right**). Pre-acquisition SEP peaks are in light blue and post-acquisition SEP peaks are in navy blue.

**Figure 4 brainsci-13-01412-f004:**
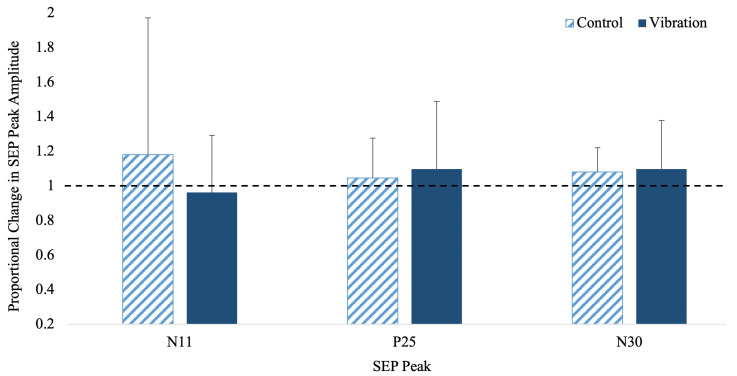
Proportional change in SEP peak amplitudes with a main effect of time: N11 peak, P25 peak, and N30 peak for controls (dashed) and vibration (solid blue). The black dotted line represents baseline values. Error bars represent SD.

**Figure 5 brainsci-13-01412-f005:**
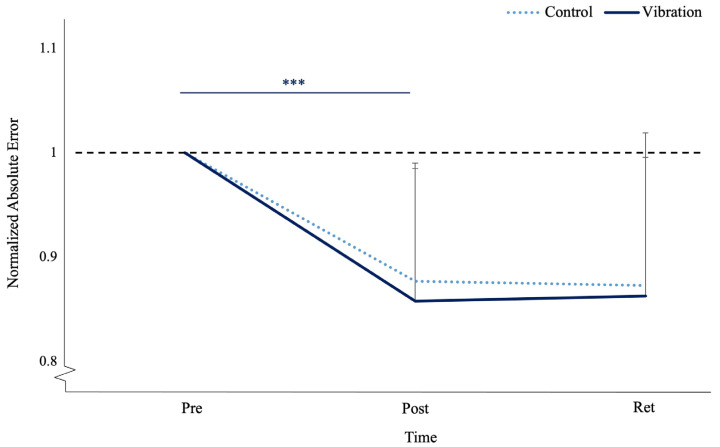
Normalized absolute error at pre-acquisition, post-acquisition, and retention of FMTT for the control (dotted) and vibration (solid blue) groups. Error bars represent SD. The black dotted line represents baseline values. (*** *p* ≤ 0.001).

**Figure 6 brainsci-13-01412-f006:**
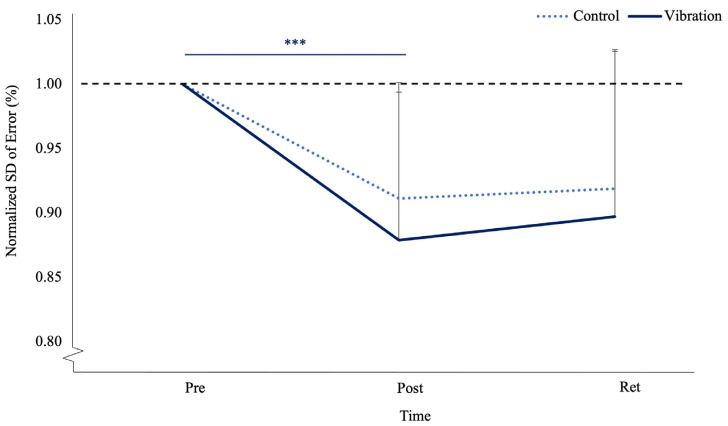
Normalized standard deviation (variance) of the error at pre-acquisition, post-acquisition, and retention of FMTT for control (dotted) and vibration (solid blue) groups. Error bars represent SD. The black dotted line represents baseline values (*** *p* ≤ 0.001).

**Table 1 brainsci-13-01412-t001:** Proportional change in SEP peak amplitudes following motor acquisition and experimental manipulation for the control and vibration groups.

		Group	
		Control	Vibration
Proportional Change in SEP Peak Amplitudes			
N9 Peak Amplitude		0.97 ± 0.10	1.03 ± 0.10
CI (95%)	Lower Bound	0.91	0.97
	Upper Bound	1.03	1.10
N11 Peak Amplitude		1.18 ± 0.79	0.96 ± 0.33
CI (95%)	Lower Bound	0.66	0.71
	Upper Bound	1.61	1.12
N13 Peak Amplitude		1.03 ± 0.31	1.06 ± 0.45
CI (95%)	Lower Bound	0.83	0.72
	Upper Bound	1.19	1.29
N18 Peak Amplitude		1.59 ± 1.07	0.78 ± 0.22
CI (95%)	Lower Bound	0.94	0.63
	Upper Bound	2.24	1.94
N20 Peak Amplitude		1.02 ± 0.29	1.21 ± 0.41
CI (95%)	Lower Bound	0.84	0.95
	Upper Bound	1.19	1.52
N24 Peak Amplitude		0.86 ± 0.28	1.16 ± 0.35
CI (95%)	Lower Bound	0.69	0.91
	Upper Bound	1.03	1.40
P25 Peak Amplitude		1.09 ± 0.24	1.19 ± 0.39
CI (95%)	Lower Bound	0.95	0.93
	Upper Bound	1.23	1.48
N30 Peak Amplitude		1.07 ± 0.14	1.11 ± 0.28
CI (95%)	Lower Bound	0.99	0.92
	Upper Bound	1.16	1.31
N60 Peak Amplitude		0.90 ± 0.35	1.00 ± 0.23
CI (95%)	Lower Bound	0.69	0.83
	Upper Bound	1.11	1.15

Values are group means ± SD for participants in control (n = 13) and vibration (n = 12) groups. The 95% confidence intervals for each SEP peak are also reported.

**Table 2 brainsci-13-01412-t002:** Absolute motor performance error and SD of motor performance error data for the control and vibration groups.

	**Time**		
	**Pre-Acquisition**	**Post-Acquisition**	**Retention**
Absolute Motor Performance Error (%)			
Control Group	0.679 ± 0.150	0.595 ± 0.115	0.592 ± 0.099
95% CI: Lower Bound	0.62	0.55	0.55
95% CI: Upper Bound	0.74	0.64	0.63
Vibration Group	0.673 ± 0.170	0.577 ± 0.097	0.581 ± 0.095
95% CI: Lower Bound	0.59	0.54	0.55
95% CI: Upper Bound	0.75	0.61	0.61
Absolute SD of Motor Performance Error (%)			
Control Group	1.23 ± 0.117	1.11 ± 0.087	1.12 ± 0.101
95% CI: Lower Bound	1.15	1.06	1.06
95% CI: Upper Bound	1.30	1.16	1.18
Vibration Group	1.22 ± 0.191	1.05 ± 0.090	1.07 ± 0.078
95% CI: Lower Bound	1.10	0.99	1.02
95% CI: Upper Bound	1.34	1.11	1.12

Values are group means ± SD for participants in control (n = 13) and vibration (n = 12) groups. The 95% confidence intervals for each measure are shown below the group means.

## Data Availability

The datasets generated during and/or analyzed during the current study are available from the corresponding author on reasonable request.

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
