# Peer review of "Neck Muscle Vibration Alters Cerebellar Processing Associated with Motor Skill Acquisition of a Proprioceptive-Based Task"

_brainsci, 2023, doi:10.3390/brainsci13101412_

Round 1

Reviewer 1 Report

This study investigated the effect of neck muscle vibration on the somatosensory evoked potentials (SEPs) modulation, particularly cerebellar processing associated component, induced by proprioceptive-related motor skill acquisition. I am interested in this study and its findings. However, there are concern points that should be updated. 

Major points

1. In this study, the authors used low-amplitude and high-frequency stimuli to elicit afferent inputs in the neck muscle, a choice made to address potential limitations in previous studies (lines 82-90). I could not understand whether this stimulus was optimal to address these issues. For instance, the authors have stated the short-lasting effect of the fatigue protocol as a limitation of previous studies, but how long did the duration of the current stimulus persist? Did this stimulus effect persist at least throughout the acquisition phase? Conversely, if the stimulus effect was too long, there is a possibility that the aftereffect of the stimulation itself may have influenced the post-SEPs. Was the aftereffect of the stimulation terminated by the time of post-SEPs recording? Furthermore, did the stimulation used in this study induce fatigue? This information should be provided by the authors.

2. In Table.2 and Figure.5, the S.D. of the PRE should be 0 as these were normalized (all values were 1). Additionally, the S.D. of normalized performance and absolute performance in Table.2 were the same. In Table. 2, the authors described "accuracy", but is this an "error"? Also, the units in Table 2 are described as "%", but is this correct?

3. There is no description of the results in terms of components such as N20 and N60, what change did they show? Information should be provided.

4. Were any instructions given to the participants during the SEPs measurement, such as attention to the stimulation and eyes opening or closing? Are the mid-latency components, which are the focus of this study, affected by attention? If so, it is important what instructions were given to control the amount of attention given by the participants.

5. Was the correlation analysis performed for all SEP components? If so, there is a risk of type 1 error due to repeated correlation analysis. The authors should consider conducting corrections for multiple comparisons.

6. Figures 6 and 7: Are the X-axis and Y-axis inverted? Both figures display the normalized absolute error on the X-axis, but the values differ between the figures. Additionally, I believe that these values are incorrect according to other figures and tables.

7. The authors addressed only the change in SEP amplitude between groups, but did the SEP amplitude actually change from PRE to POST?

8. In the FMTT, how was the range of strength of the finger press adjusted? Was it based on MVC? Please describe the range of required force during the task.

9. Did the authors analyze the change in performance during the acquisition phase? Although I understand that the post-acquisition performance did not differ between conditions, I am interested in any possible influence on the magnitude or speed of learning during acquisition.

10. In lines 195-198, the authors stated that “24-48 hours later, participants returned to complete the retention phase of the FMTT.” but why was there a large variation in the interval from post-acquisition to the retention phase? Additionally, did the authors consider the difference in this interval when doing a statistical analysis?

Minor points

11. Line 62: “due to” is repeated.

12. Please describe the interval time between blocks during the acquisition phase.

Author Response

Major points

  1. In this study, the authors used low-amplitude and high-frequency stimuli to elicit afferent inputs in the neck muscle, a choice made to address potential limitations in previous studies (lines 82-90). I could not understand whether this stimulus was optimal to address these issues. For instance, the authors have stated the short-lasting effect of the fatigue protocol as a limitation of previous studies, but how long did the duration of the current stimulus persist? Did this stimulus effect persist at least throughout the acquisition phase? Conversely, if the stimulus effect was too long, there is a possibility that the aftereffect of the stimulation itself may have influenced the post-SEPs. Was the aftereffect of the stimulation terminated by the time of post-SEPs recording? Furthermore, did the stimulation used in this study induce fatigue? This information should be provided by the authors.

The duration of the current stimulus (60Hz of vibration to the right SCM and left CEM) was 10 minutes. Based on past literature, a 60Hz vibration applied to the SCM for a duration of 10 minutes can have long-lasting effects, up to 22 hours (Pettorossi et al., 2015).

That being said, the stimulus effect does not appear to last beyond the post-acquisition phase as pilot testing using this frequency on upper limb joint position sense did not persist for more than 30 minutes. In fact, we conducted a proof of principle study to determine whether vibration alone alters SEP amplitudes or latencies, where we found that our vibration protocol had no impact on any of the SEP peaks measured. The SEP changes observed were likely the result of vibration on the acquisition of a task dependent on proprioceptive information.

Given that the stimulation frequency, the duration of stimulation presentation, and area where it was applied was small, it is unlikely that it induced neck muscle fatigue. Regardless, vibration was chosen as the experimental intervention due to its known effects on muscle spindle firing. The goals were to isolate altered neck muscle spindle input, as neck fatigue may alter the firing of pain afferents in addition to muscle spindles (Zabihhosseinian et al., 2021).

  1. In Table.2 and Figure.5, the S.D. of the PRE should be 0 as these were normalized (all values were 1). Additionally, the S.D. of normalized performance and absolute performance in Table.2 were the same. In Table. 2, the authors described "accuracy", but is this an "error"? Also, the units in Table 2 are described as "%", but is this correct?

Our apologies, this was an error. The error bars in figure 5 have also been updated in the manuscript. The results in the table are based on performance error (percent absolute error), which is the percentage of error normalized to baseline, therefore, smaller values mean better performance (or higher accuracy). We have updated the table caption and headings to say “error” instead of accuracy to minimize confusion. Additionally, the normalized motor performance error was removed from table 2, as these values are seen visually in figure 5.

  1. There is no description of the results in terms of components such as N20 and N60, what change did they show? Information should be provided.

This has now been included in the results section, as per your request. However, these have not been discussed in the discussion section as the N20 and N60 showed very minimal changes in amplitude, resulting in no statistical significance for time nor time by group interactions.

  1. Were any instructions given to the participants during the SEPs measurement, such as attention to the stimulation and eyes opening or closing? Are the mid-latency components, which are the focus of this study, affected by attention? If so, it is important what instructions were given to control the amount of attention given by the participants.

The following instructions were given to each participant: (1) be as relaxed as possible, (2) blink as you normally would, and (3) avoid unnecessary movement during the stimulation. The participants were instructed to have their eyes open during the recording as this is the standard practice (Nuwer et al., 1994). Eye blinks occurred as they normally would, which were filtered out as described in the data processing section of the methods.

Short latency SEP peaks occur before 40ms for the upper limb (Allison et al., 1991).The only mid-latency peak included in this study was the N60. It is possible this peak could be influenced by cognitive processes, however, there were no significant differences found for this peak, so the impact of attention and cognition were not discussed further. Irrespective of this fact, the pre and post SEP conditions were the same for both the vibration and control groups, as vibration was only applied in the period after SEPs and before motor acquisition. Therefore, effects of attention would have been similar between groups.

  1. Was the correlation analysis performed for all SEP components? If so, there is a risk of type 1 error due to repeated correlation analysis. The authors should consider conducting corrections for multiple comparisons.

Yes, correlation analysis was performed for all SEP components. In correcting the correlations for multiple comparisons using the benjamini-hochberg correction, the correlations were no longer statistically significant. In response to this finding, there are no correlation findings to include in the manuscript. 

  1. Figures 6 and 7: Are the X-axis and Y-axis inverted? Both figures display the normalized absolute error on the X-axis, but the values differ between the figures. Additionally, I believe that these values are incorrect according to other figures and tables.

Thank you for noticing this!   These figures have been removed from the manuscript as they were not statistically significant following the correction for multiple correlation analyses.

  1. The authors addressed only the change in SEP amplitude between groups, but did the SEP amplitude actually change from PRE to POST?

Some of SEP peak amplitudes did demonstrate a change from pre to post. We have now included the results for Main effect of time for all SEP peak amplitudes, regardless of whether they were statistically significant.

  1. In the FMTT, how was the range of strength of the finger press adjusted? Was it based on MVC? Please describe the range of required force during the task.

The force templates used for the FMTT were created to be calibrated to each participant’s average thumb MVC, 2 to 12%.

The following information has been included, see highlighted for new addition:

“The FMTT required participants to push against the force transducer using their right thumb to match a series of white-dotted square sinusoidal waves calibrated to their thumb strength, 2 to 12% of their average MVC. This range was chosen based on pilot testing, to ensure that thumb muscle fatigue did not occur during motor acquisition.”

  1. Did the authors analyze the change in performance during the acquisition phase? Although I understand that the post-acquisition performance did not differ between conditions, I am interested in any possible influence on the magnitude or speed of learning during acquisition.

Yes, this was analyzed but not reported as the focus of the work was before and after the acquisition of the motor task. That being said, we did look into the acquisition data when trying to explain the lack of differences between groups however we found that the error profile for both groups were similar. Both groups demonstrated steady improvements, at very similar rates, through the acquisition phases to post. 

  1. In lines 195-198, the authors stated that “24-48 hours later, participants returned to complete the retention phase of the FMTT.” but why was there a large variation in the interval from post-acquisition to the retention phase? Additionally, did the authors consider the difference in this interval when doing a statistical analysis?

Retention of a motor task can be assessed up to 48 hours following acquisition (ref), to assess preservation of motor performance following memory consolidation of a fast motor skill (acquired within a single training session). This time window has also been observed in various studies that used similar motor learning paradigm (Ambalavanar et al., 2022; McCracken et al., 2022; Zabihhosseinian et al., 2020)

All participants completed retention 24hrs later, hence this interval is not applicable nor needed to be considered when performing statistical analysis. Therefore, we have updated this interval to state 24 hours later in the manuscript.

Minor points

  1. Line 62: “due to” is repeated.

Thank you. This has been updated.

  1. Please describe the interval time between blocks during the acquisition phase.

The interval time between blocks is negligible, as they were administered one after another during each phase (pre-acquisition, acquisition, post-acquisition, and retention) with no rest provided between blocks. As for the acquisition phase, participants completed 3 rounds of 4 blocks (total of 12 blocks) in sequence. There was no rest between the transition of acquisition to post-acquisition.  Each phase lasted approximately 5 minutes in duration. The only reason they were divided into blocks was to allow for randomization in the order of the presentation of traces of varying levels of difficulty.

Allison, T., McCarthy, G., Wood, C. C., & Jones, S. J. (1991). Potentials evoked in human and monkey cerebral cortex by stimulation of the median nerve: a review of scalp and intracranial recordings. Brain, 114(6), 2465-2503.

Ambalavanar, U., Delfa, N. L., McCracken, H., Zabihhosseinian, M., Yielder, P., & Murphy, B. (2022). Differential changes in somatosensory evoked potentials and motor performance: pursuit movement task versus force matching tracking task. Journal of Neurophysiology, 128(6), 1453-1465.

McCracken, H., Murphy, B., Ambalavanar, U., Zabihhosseinian, M., & Yielder, P. (2022). Sensorimotor Integration and Motor Learning During a Novel Visuomotor Tracing Task in Young Adults with Attention-Deficit/Hyperactivity Disorder. Journal of Neurophysiology.

Nuwer, M. R., Aminoff, M., Desmedt, J., Eisen, A. A., Goodin, D., Matsuoka, S., . . . Vibert, J.-F. (1994). IFCN recommended standards for short latency somatosensory evoked potentials. Report of an IFCN committee. Electroencephalography and clinical Neurophysiology, 91(1), 6-11.

Pettorossi, V. E., Panichi, R., Botti, F. M., Biscarini, A., Filippi, G. M., & Schieppati, M. (2015). Long-lasting effects of neck muscle vibration and contraction on self-motion perception of vestibular origin. Clinical Neurophysiology, 126(10), 1886-1900.

Zabihhosseinian, M., Yielder, P., Berkers, V., Ambalavanar, U., Holmes, M., & Murphy, B. (2020). Neck muscle fatigue impacts plasticity and sensorimotor integration in cerebellum and motor cortex in response to novel motor skill acquisition. Journal of Neurophysiology, 124(3), 844-855.

Zabihhosseinian, M., Yielder, P., Wise, R., Holmes, M., & Murphy, B. (2021). Effect of Neck Muscle Fatigue on Hand Muscle Motor Performance and Early Somatosensory Evoked Potentials. Brain Sciences, 11(11), 1481.

Reviewer 2 Report

This study investigates the impact of Neck Muscle Vibration on Cerebellar Processing in relation to Motor Skill Acquisition of a Proprioceptive-Based Task. The study involved the enrolment of 25 right-handed participants who underwent electrical stimulation over the right median nerve to elicit short and middle-latency somatosensory evoked potentials (SEPs) both before and after acquiring a force-matching tracking task. The participants were divided into two groups: the control group, who received 10 minutes of rest, and the vibration group, who received 10 minutes of 60Hz vibration applied to the right sternocleidomastoid and left cervical extensors. Task performance was evaluated 24 hours later to assess retention. Notable findings include significant interactions between time and group for the N18 SEP peak, with a 21.77% decrease in the VIB group compared to a 58.74% increase in the CONT group, and the N24 SEP peak, showing a 16.31% increase in the VIB group compared to a 14.05% decrease in the CONT group. Both groups exhibited improvements in motor performance post-acquisition and at retention. These group-dependent changes in SEP peaks, associated with cerebellar input (N18) and cerebellar processing (N24), suggest that altered proprioceptive input resulting from neck vibration impacts cerebellar pathways. I have a few comments that should be considered.

Specific comments:

1)      The most significant issue here is the absence of an adequate control condition. The CONT group was simply kept at rest without receiving any stimulation. This methodological choice is particularly critical as it leaves the study participants non-blinded to the experimental conditions. In my opinion, it is essential to add a control experiment by applying vibration to another part of the body. This would allow to demonstrate that the observed results described in the study are specifically dependent on the stimulation of the neck muscles.

2)      Consider shortening the introduction to focus on the study's rationale and enhance the background.

3)      On page 3, line 100, when referencing the study on neck vibration's impact on upper limb proprioception, please provide the appropriate reference for citation.

4)      On page 6, in the paragraph concerning Statistical Analysis (2.8), it's essential to specify whether a correction was applied to multiple correlations. Please clarify this aspect for the readers.

5)      On page 13, lines 434 and 435, where it's mentioned that motor learning can be assessed behaviourally by measuring improvements in performance accuracy after acquiring a novel motor skill, consider acknowledging that motor learning can also be quantified as improved motor performance, not necessarily accuracy. Jus as one example, see Muellbacher W, Ziemann U, Wissel J, Dang N, Kofler M, Facchini S, Boroojerdi B, Poewe W, Hallett M. Early consolidation in human primary motor cortex. Nature. 2002 Feb 7;415(6872):640-4. doi: 10.1038/nature712. Epub 2002 Jan 23. PMID: 11807497. Please consider and discuss this issue adequately in the paper.

6)      Regarding Figure 2 and Figure 4, consider potential improvements in their layout and colour schemes. Additionally, address the issue of disproportionately low error bars in the 'pre' condition of Figure 4 and provide an explanation for this observation. This will help enhance the visual presentation and comprehensibility of the figures.

Author Response

Specific comments:

  • The most significant issue here is the absence of an adequate control condition. The CONT group was simply kept at rest without receiving any stimulation. This methodological choice is particularly critical as it leaves the study participants non-blinded to the experimental conditions. In my opinion, it is essential to add a control experiment by applying vibration to another part of the body. This would allow to demonstrate that the observed results described in the study are specifically dependent on the stimulation of the neck muscles.

Thank you for providing your insight on the lack of adequate control condition.

While the vibrators weren’t turned on, the participants were unaware that they were vibrators at all. They were told that electrodes were being placed on their neck for 10 minutes while they rested between sessions. Additionally, the SEPs recorded are pre-cognitive, and as such, it cannot be influenced by information provided to our participants. We also completed a proof-of-concept pilot to examine if the effects of vibration (without the motor learning task) lead to changes in SEPs and found no changes in SEP peaks from pre to post vibration.

In light of this comment, we have updated the limitations section to include this as a limitation of this work, which will be addressed in future studies.

  • Consider shortening the introduction to focus on the study's rationale and enhance the background.

This has been done.

  • On page 3, line 100, when referencing the study on neck vibration's impact on upper limb proprioception, please provide the appropriate reference for citation.

Thank you, we have updated the reference accordingly.

  • On page 6, in the paragraph concerning Statistical Analysis (2.8), it's essential to specify whether a correction was applied to multiple correlations. Please clarify this aspect for the readers.

Correction of multiple correlations was performed using the benjamini-hochberg correction, revealing that none of the correlations were statistically significant. In response to this finding, it has been omitted from the manuscript. 

  • On page 13, lines 434 and 435, where it's mentioned that motor learning can be assessed behaviourally by measuring improvements in performance accuracy after acquiring a novel motor skill, consider acknowledging that motor learning can also be quantified as improved motor performance, not necessarily accuracy. Just as one example, see Muellbacher W, Ziemann U, Wissel J, Dang N, Kofler M, Facchini S, Boroojerdi B, Poewe W, Hallett M. Early consolidation in human primary motor cortex. Nature. 2002 Feb 7;415(6872):640-4. doi: 10.1038/nature712. Epub 2002 Jan 23. PMID: 11807497. Please consider and discuss this issue adequately in the paper.

The discussion of motor behavior has been corrected to consider both motor performance and performance accuracy.

  • Regarding Figure 2 and Figure 4, consider potential improvements in their layout and colour schemes. Additionally, address the issue of disproportionately low error bars in the 'pre' condition of Figure 4 and provide an explanation for this observation. This will help enhance the visual presentation and comprehensibility of the figures.

Figure 4 is reflective of the proportional change in SEP peak amplitude, which has been made clear in the figure caption. The “pre” condition is the normalized SEP peak amplitude (mean = 1; SD = 0), seen as the low error bars in the pre-condition. We have updated all the colour schemes and layout of these figures in the manuscript.

Reviewer 3 Report

In my opinion, control setting in this study is not appropriate. Vibration application on the neck does not only make effects on the neck muscle such as SCM. Pain sensation might be also involved. However, turn-off group is enrolled as control group in this study.

1. I have small questions about the control group setting. How about if the vibration apply to the left SCM and right CEM?

2. ”25” at the beginning of some sentences should changed into "Twenty-five".

Author Response

In my opinion, control setting in this study is not appropriate. Vibration application on the neck does not only make effects on the neck muscle such as SCM. Pain sensation might be also involved. However, turn-off group is enrolled as control group in this study.

The chosen vibration frequency (60Hz) has been known to specifically target muscle spindles, without evoking the pain receptors (c-fibers) within the neck musculature (Brown, Engberg et al. 1967). During the piloting phase of this work, participants were asked to report whether they experienced somatosensations with respect to vestibular system and/or nociceptive system (e.g. dizziness, nausea, pain, etc.), where none of the participants reported such thing, except for the sensation of a movement illusion. In order to state with certainty that the changes observed between groups were the result of the neck muscle vibration, a control group who had vibrators placed over the right SCM and left CEM, that were turned-off was appropriate for this study, which was particularly aimed at investigating the effect of altered muscle spindle firing from the neck muscles on upper limb somatosensory processing.

  1. I have small questions about the control group setting. How about if the vibration apply to the left SCM and right CEM?

As mentioned, the control group had the vibrators placed on their right SCM and left CEM muscles without them being turned-on.  The purpose of this was to control for the weight of the vibrators themselves, and any skin sensations induced by the presence of the vibrators which could have resulted in habituation.

Although this question is beyond the scope of the current study, previous work on neck vibration by Guerraz et al. suggested the following:

Previous work has examined the effects of vibration on SCM and contralateral CEM on both sides of the neck (Knox, Cordo et al. 2006). Additionally, other work examining how changes in head position impacted proprioception and found that left head rotation elicited a more dramatic shift in proprioceptive accuracy (Guerraz, Caudron et al. 2011). Therefore, we chose to stimulate the right SCM and left CEM in an attempt to elicit a movement illusion of left head rotation. Other work has shown that right rotation elicited a more robust change in proprioceptive accuracy but in either case, rotation of the head in either direction increased error of the upper limb.

  1. ”25” at the beginning of some sentences should changed into "Twenty-five".

Thank you for bringing this to our attention. We have changed this throughout the manuscript.

Brown, M., I. Engberg and P. Matthews (1967). "The relative sensitivity to vibration of muscle receptors of the cat." The Journal of physiology 192(3): 773-800.

Guerraz, M., S. Caudron, N. Thomassin and J. Blouin (2011). "Influence of head orientation on visually and memory-guided arm movements." Acta psychologica 136(3): 390-398.

Knox, J., P. Cordo, R. Skoss, S. Durrant and P. Hodges (2006). "Illusory changes in head position induced by neck muscle vibration can alter the perception of elbow position." Behavioral neuroscience 120(6): 1211.

Reviewer 4 Report

starting line 68- Numbered "N" may be of limited understanding for not specialized readers; we suggest some information about or even a picture of their locations in a experimental setting. (ok, is partially explained in the discussion)

starting line 78- Muscle spindles also have the capability to participate in the tonus modulation; maybe, giving some more information about such complex muscle brain could help to understand complexity of bottom processes also involved (and not only top ones).

lines 131-134- include references to support exclusion criteria

lines 219-223:

1- Explain clearly the advantadge of percent error (e.g., individual differences).

2- Why was not estimated variable error, or other type of errors (e.g., total error)? Each type of error can give you specific information, that may help to better are more profoundly understand motor behaviour, and consequences (e.g., if movement becames more variable with fatigue and pain, precision movements will be less efficient). Because your study is based in precision movements, variable error seems to be even more pertinent. Any book of motor controland learning will give you pratical formulas to obtain results about these type of errors and what they mean in terms of motor movement.

lines 251-254- 95% Confidence intervals must be calculated and presented, preserving respective signs. Mandatory.

Lines 260-262- select and include a good reference to rehinforce importance of "N9 SEP peak pre to post"

Table 1- Do you think that coeficient of variance (and variable error...) could contribute for results discussion? It seems higher (greater SD) in N18 Control Group compared to Vibration one... Notice your reflections (lines 357-371), it may be that less capability to control fine movements would result in freezing degrees of freedom (Bernstein Problem), hypothesis that can be also tested with statistical correlation.

line 397- paragraph

lines 420-422- It would be interesting to include a transfer test for both groups. If vibration one needed more plasticity, their adaptability to a new condition should be greater (without vibration, of course) (see also lines 447-453, we totally agree, augmented difficulty in acquisition phase may favor retention and transfer, e.g., contextual interference hypothesis)

lines 471-476- And what about muscle spindles? Bottom-up control instead of a top-down one?

4.4. Limitations- Movement analyses (process) (e.g., using IMUs) instead of absolut error (product)

3.3. Pearson’s Correlations- calculate and present confidence interval values, preserving respective signs. Mandatory.

Thanks for your study 

Author Response

  1. starting line 68- Numbered "N" may be of limited understanding for not specialized readers; we suggest some information about or even a picture of their locations in an experimental setting. (ok, is partially explained in the discussion)

Additional information has been provided for the SEP Peaks that were discussed, to provide some context for readers.

  1. starting line 78- Muscle spindles also have the capability to participate in the tonus modulation; maybe, giving some more information about such complex muscle brain could help to understand complexity of bottom processes also involved (and not only top ones).

We have added the following line:

 “In addition, altered spindle firing could possibly, impact neck muscle tone which could have kinetic chain effects on upper limb muscles.”

  1. lines 131-134- include references to support exclusion criteria

References have been included, as per your suggestion.

  1. Explain clearly the advantage of percent error (e.g., individual differences).

Percent error permits for the assessment of individual improvement with respect to motor performance, and is commonly used method to assess motor learning related behavioral changes. It provides information regarding motor performance without the consideration of movement strategies, and variability, etc. The challenge with movement strategies and variability is that they differ so much between individuals that it makes comparison difficult.

  1. Why was not estimated variable error, or other type of errors (e.g., total error)? Each type of error can give you specific information, that may help to better are more profoundly understand motor behaviour, and consequences (e.g., if movement becomes more variable with fatigue and pain, precision movements will be less efficient). Because your study is based in precision movements, variable error seems to be even more pertinent. Any book of motor control and learning will give you practical formulas to obtain results about these types of errors and what they mean in terms of motor movement.

We have now included the standard deviation of motor performance error as well.

  1. lines 251-254- 95% Confidence intervals must be calculated and presented, preserving respective signs. Mandatory.

95% confidence intervals have been included.

  1. Lines 260-262- select and include a good reference to reinforce importance of "N9 SEP peak pre to post"

The reference has been included.

  1. Table 1- Do you think that coefficient of variance (and variable error...) could contribute for results discussion? It seems higher (greater SD) in N18 Control Group compared to Vibration one... Notice your reflections (lines 357-371), it may be that less capability to control fine movements would result in freezing degrees of freedom (Bernstein Problem), hypothesis that can be also tested with statistical correlation.

A correlation analysis between the normalized N18 SEP peak and normalized motor performance at post-acquisition was performed, which did not yield statistical significance. This suggests that improvements in motor performance did not correspond with subsequent decreases in olivary-cerebellar inhibition.

We have added a line to the discussion in response to your comment, which states the following: “that vibration may have impacted the flexibility of the CNS to respond through the freezing of various motor systems to achieve the motor skill, as seen in the decreased variability of the N18 SEP peak.”

  1. lines 420-422- It would be interesting to include a transfer test for both groups. If vibration one needed more plasticity, their adaptability to a new condition should be greater (without vibration, of course) (see also lines 447-453, we totally agree, augmented difficulty in acquisition phase may favor retention and transfer, e.g., contextual interference hypothesis)

Thank you for bringing this up. We have added this as a future direction. Indeed, we do intend to address this in subsequent studies.

  1. lines 471-476- And what about muscle spindles? Bottom-up control instead of a top-down one?

We have addressed this comment in the introduction as information pertaining to the correlations were removed, since nothing was statistically significant following corrections for multiple correlation comparisons.

  1. Limitations- Movement analyses (process) (e.g., using IMUs) instead of absolute error (product)

Thank you for providing this perspective regards to the motor behavior; this has been added as a future direction, as it was beyond the scope of the current study.  The work at hand was to examine differential changes in SEP peak amplitude in response to acquisition of a motor following neck muscle vibration. Motor performance was also reported, to determine motor learning related changes. The examination of movement analyses would not have yielded usable information in the current study, as we did not control for movement strategies. Interestingly, in our related work, we did show altered upper limb proprioception (Tabbert, Ambalavanar et al. 2022) which suggests that movement trajectories might be impacted as well and this will be addressed in our future work.

  1. Pearson’s Correlations- calculate and present confidence interval values, preserving respective signs. Mandatory.

This is no longer relevant as the assessment of Pearson’s correlations has been removed from the manuscript.

Tabbert, H., U. Ambalavanar and B. Murphy (2022). "Neck Muscle Vibration Alters Upper Limb Proprioception as Demonstrated by Changes in Accuracy and Precision during an Elbow Repositioning Task." Brain Sciences 12(11): 1532.

Round 2

Reviewer 1 Report

In response to my previous comments, the authors have undertaken adequate revisions to the manuscript. Consequently, no further points of contention or inquiry arise on my part.

Author Response

Thank you for your feedback.

Reviewer 2 Report

The authors have addressed my comments. I have no further requests.

Author Response

Thank you for your feedback.